# TLR9 and Glioma: Friends or Foes?

**DOI:** 10.3390/cells12010152

**Published:** 2022-12-30

**Authors:** Emna Fehri, Emna Ennaifer, Rahima Bel Haj Rhouma, Monia Ardhaoui, Samir Boubaker

**Affiliations:** 1HPV Unit Research, Laboratory of Molecular Epidemiology and Experimental Pathology Applied to Infectious Diseases, Pasteur Institute of Tunis, Tunis 1002, Tunisia; 2Department of Human and Experimental Pathology, Pasteur Institute of Tunis, Tunis 1002, Tunisia

**Keywords:** TLR9, glioma, tumor regression, tumor progression, dichotomic role

## Abstract

Toll-like receptor 9 (TLR9) is an intracellular innate immunity receptor that plays a vital role in chronic inflammation and in recognizing pathogenic and self-DNA in immune complexes. This activation of intracellular signaling leads to the transcription of either immune-related or malignancy genes through specific transcription factors. Thus, it has been hypothesized that TLR9 may cause glioma. This article reviews the roles of TLR9 in the pathogenesis of glioma and its related signaling molecules in either defending or promoting glioma. TLR9 mediates the invasion-induced hypoxia of brain cancer cells by the activation of matrix metalloproteinases (2, 9, and 13) in brain tissues. In contrast, the combination of the TLR9 agonist CpG ODN to radiotherapy boosts the role of T cells in antitumor effects. The TLR9 agonist CpG ODN 107 also enhances the radiosensitivity of human glioma U87 cells by blocking tumor angiogenesis. CpG enhances apoptosis in vitro and in vivo. Furthermore, it can enhance the antigen-presenting capacity of microglia, switch immune response toward CD8 T cells, and reduce the number of CD4CD25 Treg cells. CpG ODN shows promise as a potent immunotherapeutic drug against cancer, but specific cautions should be taken when activating TLR9, especially in the case of glioblastoma.

## 1. Introduction

Gliomas are the most prevalent and lethal type of brain tumor [1]. The high incidence and high mortality rates observed for gliomas, predominantly in developed countries, are presumably due to an underestimation of glioma cases elsewhere and restricted facilities for medical care [2,3,4]. The median survival time is 15 months [2]. Treatment is complex, and the first-line treatment is aggressive and entails surgery, radiation therapy, and chemotherapy [5]. Unfortunately, these therapies provide limited overall survival benefits and are far from satisfactory. Thus, alternative novel therapies that enhance or work in cooperation with conventional treatments are urgently required. Glioma carcinogenesis is a multistage process that involves complicated mechanisms. Many factors, such as chronic inflammation, are inextricably involved in the pathogenesis and progression of glioma. Glioma cells secrete specific cytokines, thus creating an immunosuppressive environment involving signal transducer and activator of transcription 3 (STAT3) signaling, which inhibits immune cells [6,7,8]. However, the main mechanisms underlying glioma progression remain unclear.

Toll-like receptors (TLRs) recognize particular microbial structures [9]. Toll-like receptor 9 (TLR9) is stimulated by unmethylated cytosine-guanine nucleotide sequences (CpGs) in bacterial, viral, and fungal DNA fragments, as well as synthetic oligonucleotides [10] or self-DNA [11]. It has been previously reported that TLR9 is only expressed in some immune cells, such as plasmacytoid dendritic cells (pDC), and acts as a potent inducer of Interferon (IFN) secretion to overcome viral infection. However, recent studies, including ours, have focused on their expression in tumor cells, such as cervical cancer [12,13,14], postulating their role in carcinogenesis. TLR9’s function in tumorigenesis is also a double-edged sword, and although it has a potential anti-cancer capacity, a pro-tumorigenic relevance is more likely [15,16].

It is therefore necessary to conduct large-scale studies to assess the significance of these receptors in specific neoplastic cells. This review focuses on glioma and may reveal new strategies that would selectively drawback tumor cells and deflect host antitumor responses regarding the eradication of these deadly cancers and could pave the way for the development of new immunotherapeutic targets. Here, we hypothesized that TLR9 might play dual roles in cancers such as glioma, and we review and summarize the current knowledge about TLR9-signaling in the pathogenesis of glioma.

## 2. Glioma

Gliomas, the most recurrent type of primary brain cancer, arise from the glial tissue of the central nervous system (CNS). The adult CNS comprises three types of glial cells that give rise to astrocytes, oligodendrocytes, and ependymal cells. The most frequent tumors are astrocytomas, originating from astrocytes or their precursors. Established on morphology and malignant behavior features, and as reported by the World Health Organization (WHO) criteria, glioma tumors are histologically divided into Grades I and IV. Malignant gliomas constitute a range of clinicopathological features, from low-to-high-grade malignancies; roughly all low-grade tumors ultimately progress to high-grade malignancy. Grade I tumors are commonly benign and have a good prognosis, and radical surgical excision generally results in healing. These tumors regularly appear in children [17,18], and histological examination of Grade II tumors is marked by hypercellularity and has a median survival of approximately 5–8-year median survival [19,20]. Grade II tumors show a proportion of recurrence and an increase in grade. Grade III gliomas exhibit hypercellularity, nuclear atypia, prominent mitotic figures, and a 3-year median survival [21]. Glioblastomas (GBMs) are the most common and detrimental cancers among a broad range of glial brain tumors. The histological characteristics of Grade IV gliomas include hypercellularity, nuclear atypia, mitotic figures, and angiogenic and/or necrotic features. GBM is intrusive and can be diagnosed atrociously. Chemotherapy and irradiation are far from potent, with an average survival time of approximately 15 months.

It is now fully understood that the knowledge of molecular alterations has greatly expanded. In the WHO 2016 revised reclassification, molecular markers are increasingly and newly included in combination with a classical histological cancer diagnosis. Isocitrate dehydrogenase (IDH) mutation and 1p/19q co-deletion are the most common genetic alterations analyzed in these tumors [22]. Moreover, comparative genomic research has shown that gains or losses of several chromosomes containing either oncogenes or tumor suppressor genes are common to all GBM tumors and stratified GBM into many subtypes [23]. However, the major roles of immune-associated molecules in the overthrow or progression of glioma are yet to be fully elucidated.

## 3. TLR9 Overview

### 3.1. TLR9 Discovery, Structure, Ligands 

TLR9 is an intracellular endosomal receptor. It was first cloned in 2000, and its gene is located on 3p21.3 [24]. This receptor is involved in the recognition of unmethylated CpG-DNA of bacterial, viral, and parasitic origin [25], but also of self-DNA in immune complexes [26]. TLR9 ligands, B/K-type and A/D-type, are two subtypes of CpG oligodeoxynucleotides (CpG-ODNs). The pure phosphorothioate backbone B/K-type CpG-ODNs lead to B cell proliferation and dendritic cell (DC) maturation. TLR9 is first expressed by B cells and plasmacytoid DCs (pDCs) [27]. TLR9 is a type I transmembrane protein composed of three domains: an N-terminal extracellular domain composed of leucine-rich repeats (LRRs), a hydrophobic transmembrane domain, and a cytoplasmic Toll/interleukin 1 receptor (TIR) domain [28].

TLR9 is an intracellular receptor and its cytoplasmic localization is required to overcome self-host nucleic recognition and avoid autoimmune diseases [29]. The capacity to distinguish between self- and non-self-nucleic acids relies on molecular recognition. Instead, the capacity to bind nucleic acids has been tightly linked to a unique localization and regulatory program. TLR9 is synthesized and localized in the endoplasmic reticulum; they translocate to the endosome upon ligands stimulation. Several key molecules are required for the trafficking and activation of nucleic acid-sensing toll-like receptors in endosomes, such as the transmembrane protein: the uncoordinated 93 homolog B1 (UNC93B1). TLR9 interacts with UNC93B in the endoplasmic reticulum, facilitates loading into COPII vesicles, and is transported to the Golgi [30,31]. UNC93B1 and TLR9 traffic together to the surface and, through its recruitment of AP-2-mediated internalization, reach endocytic compartments [32]. Upon the entry of TLRs into the endocytic pathway, additional sorting to specific signaling compartments, from which IRFs or NF-κB can be activated, is required.

Further compartmental specialization is generated by adapter-related protein complex-3 (AP-3), which interacts with TLR9 and directs the receptor to the endosomal compartments dedicated to type I interferon signaling [33]. The brain and DC-associated LAMP-like molecules (BAD-LAMP/LAMP5) control TLR9 trafficking to LAMP1+ late endosomes in human plasmacytoid dendritic cells (pDC), leading to NF-κB activation and tumor necrosis factor (TNF) production upon DNA detection [34].

Other accessory molecules are required for transport trafficking and activation of nucleic acid-sensing toll-like receptors in endosomes; the master chaperone 96 kDa glycoprotein (gp96), the solute carrier protein superfamily member Slc15a4, and the proteins associated with the toll-like receptor (TLR) 4, PRAT4A [35,36,37].

The ectodomain of toll-like receptor 9 is cleaved to generate a functional receptor. TLR9 requires compartmentalized proteolytic processing to initiate the signaling process upon translocation to the endolysosomes. The cleavage site in the ectodomain of TLR9 is likely in the region encompassing amino acids 378–475 of TLR9 receptor requiring asparagine endopeptidase (AEP) or cathepsin family members. General regulatory strategy for all TLRs involved in nucleic acid recognition. TLR9 contains a large non-conserved Z-loop between LRR14 and LRR15 that is susceptible to cathepsin-mediated proteolysis [38,39,40] (Figure 1).

### 3.2. TLR9 Signaling Pathway 

Once the ligand binds to the cleaved TLR9 receptor, the TLR9/ligand complex activates a downstream pathway that initiates the activation of various intracellular signaling molecules and transcription factors to elicit an immune response against the recognized pathogen [9]. The TIR domains of TLR9 and MYD88 interact. Activated MYD88 interacts with interleukin-1 receptor-associated kinase (IRAK1) and IRAK4 via its death domain, contributing to the involvement of TNF receptor-associated factor (TRAF3) and TRAF6. The activation of the TLR9-IRAK-TRAF6 signaling pathway triggers the stimulation of transcription factors such as mitogen-activated protein kinase (MAPK), activator protein 1 (AP-1), and the nuclear factor NF-κB. NF-κB phosphorylation initiates the activation of several genes, including cytokines, chemokines, addressing molecules, and costimulatory molecules such as CD80 and CD86 [41]. TLR9-IRAK1-TRAF3 signaling cascade leads to type I interferon stimulation by activating transcription factor interferon regulatory factor 7 (IRF7). TLR9 stimulation also induces the activation of natural killer cells, T cells, B cells, and pDCs, thereby enhancing pro-inflammatory and T Helper 1 (Th1) cytokines such as IL12 stimulation and CTL cytotoxicity capable of eliminating viral pathogens and cancer [42].

### 3.3. TLR9 Expression on Immune Cells

In humans, the expression of TLRs depends on the immune cell type. Among human antigen-presenting cells (APCs), TLR9 is expressed predominantly on plasmacytoid dendritic cells but not on myeloid-derived DCs (mDCs) [43] monocytes and macrophages also express TLR9 [44]. In the human adaptive immune system, TLR9 is expressed in activated T cells and memory B cells [45,46].

In B-cells, CpG-ODNs induce secretion of several cytokines such as IL6, IL10 CCL3 (MIP1α), and CCL4 (MIP1β), preventing apoptosis triggered by surface antigen-receptor cross-linking or other apoptotic agents). CpGs were reported to upregulate TRAIL on B cells in PBMC, thereby enhancing their ability to kill tumor cells and promote Ig secretion and TLR expression on immune cells, which generally supports the therapeutic purpose of its ligands [47,48,49].

### 3.4. TLR9 Critically Bridges Innate and Adaptative Immunity: How Does the TLR9-MyD88 Pathway Promote Adaptive Immune Responses?

Research conducted on TLR9-deficient mice demonstrated the role of TLR9 not only in the secretion of pro-inflammatory cytokines but also in the induction of CD4 T helper 1 (Th1)-biased immune response and the proliferation of B cells [24,50].

Indeed, infection model studies support the crucial role of the TLR9-MyD88 pathway in the induction of adaptive immune responses to infections, such as in fighting herpes simplex virus 1 and 2, murine cytomegalovirus, and adenovirus infections. TLR9 can also recognize bacterial DNA, such as Mycobacterium tuberculosis, Brucella, Streptococcus pneumoniae, Helicobacter, and Cryptococcus neoformans [51,52,53,54,55,56].

The activation of adaptive immunity is dictated by the activation of innate immunity. Adaptive immunity is characterized by the clonal expansion of antigen-specific T and B cells. Dendritic cells (DCs) are cells of the innate immune system that bridge the innate and adaptive immune responses [57].

TLRs expressed on DCs are critically involved in the maturation of immune cells and initiate the activation of adaptive immunity [58]. Upon DC stimulation by CpG-DNA, it produces pro-inflammatory cytokines such as TNF-α/β, IL-6, and IL-12, and upregulates the surface expression of costimulatory molecules such as CD40, CD70, B7-1 (CD80), B7-2 (CD86), and major histocompatibility complex (MHC) class II. Upon antigen capture, immature DCs become licensed and activate naïve T cells [59,60]. T-cell activation and differentiation into effector cells occur after naïve T cells receive multiple signals from DCs, including antigen presentation via the T-cell receptor (TCR) of naïve T cells through CMHII, co-stimulation (signals 1 and 2), and other signals such as the cytokine milieu promoting the differentiation of T lymphocytes into cytotoxic effector cells. DCs are implicated in the cross-presentation of MHC-I molecules. CD4 T-cell engagement induces surface expression of the CD40 ligand, stimulating CD40 signaling in DC [61,62]. The costimulatory molecule CD70 is also implicated in the priming of CD8 T cells upon DC-TLR activation [63].

In addition, upon TCR ligation, CpG-ODNs can induce IL2 receptor and IL2 secretion and increase the cytolytic activity of T cells [64]. NK cells are strongly activated by CpG-ODNs. NK cell activation depends on the secretion of cytokines by DCs, which are IL12, TNF, and IFN. IFNs are primordial for the induction of an efficient immune response to tumors [65]. Therefore, the activation of the TLR-IFN type I signaling pathway is of therapeutic importance because it eliminates DC-induced tolerance and generates an antitumor response. Additionally, DCs activated by TLRs can mediate antitumor responses by presenting antigens, thereby initiating a T-cell response and inducing cytotoxicity in tumor cells. IFN can regulate the functions of natural killer cells (NK) and is very important for the modulation of tumor growth [66]. Furthermore, TLR9 agonists can exert antitumor effects by suppressing angiogenesis, blocking tumor growth through cell cycle arrest, or inducing autophagy. TLR-induced interferon plays an important role because it reduces angiogenesis and metastasis [67].

Depending on the context of Ag presentation, pDCs lead to immunity by stimulating T-cell priming or promoting the induction of T-cell tolerance. pDCs can promote tolerance by presenting antigens to CD4^+^ T cells and inhibiting their activation or inducing regulatory T cells (Tregs), which promotes tumor progression in several solid tumors [68,69] (Figure 2).

### 3.5. The Expression of TLR9 in Cancer Cells Can Corrupt the Process

TLR9 stimulation demonstrated TLR9/AP1/cyclin D1 signaling-mediated carcinogenesis in oral squamous cell carcinomas (HB cells in vitro) [70,71]. In addition, the CXCR4/SDF-1/Akt pathway is essential for the TLR9 pathway to enhance the metastasis of lung cancer cells in vitro (95D cells) [72]. CpG ODN led to the activation of NF-κB and enhanced expression of matrix metalloproteinase (MMP)-2, MMP-7, and cyclooxygenase-2 (COX-2) mRNA TLR9 in esophageal cancer suggesting its role in cell proliferation and differentiation [73]. *H. pylori* acts through TLR2/9 to activate the PI-PLCλ/PKCα/c-Src/IKKα/β and NIK/IKKα/β pathways, resulting in NF-κB and Cox2 expression. The Cox2 expression may also contribute to gastric carcinogenesis [74,75].

TLR9 exerts various pro-cancerous effects. The matrix metalloproteinase (MMP) 2/9 -TLR9 axis is of particular interest because most cancerous cells secrete massive quantities of MMP2/9. MMP-2 and MMP-9, also known as Gelatinase A and B, respectively, play critical roles in tumor cell invasion and metastasis, as they can degrade the major components of the extracellular matrix (ECM), a major component of the tumor microenvironment. MMP-2 and MMP-9 are thought to be the key enzymes in this process because they degrade type IV collagen [76,77]. TLR9-MMP signaling regulates invasion in a variety of cancer cells, including breast and prostate cancers [78,79], and the TLR-9-mediated invasion of oral cancer cells is promoted via activation of the DNA-binding activity of at least in part AP-1 TLR-9 signaling [71]. In brain tissues, hypoxia has also been shown to activate MMP-2, -9, and -13. These proteases may be TLR9-regulated in brain cancer cells. These observations need further analysis [80]. The GL261 glioma cell line and activation of TLR2 upregulated the expression of MMP2 and MMP9 to promote tumor invasion, indicating that TLR2 signaling in glioblastoma stem cells (GSCs) is involved in the invasiveness of glioma. However, TLR2 upregulated MMP2 and MMP9 expression in GL261 glioma cell lines and promoted tumor invasion, indicating that TLR2 signaling in GSCs is involved in the invasiveness of glioma [81].

## 4. TLR9 Expressions and Function in Gliomas

In the CNS, TLR9 is expressed in neurons, glial cells, and immune cells [82]. TLRs play important roles in cancer cells and the modulation of immune responses in glioma. Upon ligand recognition, TLR9 activation promotes downstream signaling, supporting either tumor progression or suppression, and therefore, can be used as a potential target in cancer therapy [83,84,85].

TLR9 was reported to be expressed in human glioma cell lines U251 and U87, the murine cell line C6 primary human glioma biopsies, and isolated GSCs [86,87,88,89].

### TLR Expressions in the Glioma Microenvironment

Microglia are tissue-resident macrophages of myeloid origin that are essential for brain-specific immune surveillance. TLR 9 is highly expressed in both human microglia in the normal brain parenchyma and tumor-infiltrating microglia [90]. In response to pathogen attack or other pro-inflammatory stimuli, the expression of microglial TLRs is responsible for the innate immune system of the brain. Microglial infiltration is highly influenced by the tumor microenvironment. Synergistic activation of TLR3 and TLR9 in microglia reinforces the secretion of pro-inflammatory factors, phagocytic activity, and suppression of glioma growth [91].

Glioma cells exhibit stem cell-like phenotypes called glioma stem cells (GSC), which are known to be aggressive and resistant to therapy. CpG-ODN was observed to activate TLR9 to promote the growth of GSCs through the activation of signal transducer and activator of transcription 3 (STAT3) signaling in cultured cells; silencing TLR9 expression abrogated the GSC development [92].

In the human DC infiltration distribution, there were fewer pDCs in GBM specimens than in normal brains. Grade III and IV malignant gliomas are associated with a potent immunosuppressive tumor microenvironment that escapes the host antitumor response. One of the characteristic features of glioma is immunosuppression in the presence of Tregs in the immunosuppressive glioma microenvironment, which is potentiated by the suppression of APC functions via the expression of immunosuppressive cytokines, such as IL-10 and TGF-β, contributing to the abolition of effector T-cell progression in the murine model host pDCs promoting glioma [93].

## 5. Current Status of TLR9 Agonists in Glioma Treatment and Clinical Trial Based on CpG Agonists

The expression pattern profile and signaling mechanisms of TLRs make them potential targets for glioma therapy, where multiple routes may be targeted to aid the development of effective clinical strategies. An important aspect of the utilization of TLRs in glioma clinical trials is the application of TLR agonists as single agents to suppress tumors.

TLR agonists have been reported to initiate or suppress immune responses in the glioma environment upon binding to specific TLRs. CpGODNs can be administered subcutaneously, intrathecally, or intracranially. Local administration of TLR agonists is of particular interest in immunotherapies [84,94].

In this regard, several clinical phase studies have been carried out and others are currently in process; however, the results are controversial and have not led to a definitive position regarding the use of TLR agonists as adjuvant therapy to treat CNS tumors.

Phase I clinical studies were conducted on CpG -28 (sequence 5′-TAAACGTTATAACGTTATGACGTCAT-3′) administered locally to 24 patients with recurrent GBM by convection-enhanced delivery. The dose of CpG-28 was escalated from 0.5 mg to 20 mg dose level. Two patients whose tumors were growing at the time of inclusion showed a minor response (29% and 20% reduction, respectively, in the product of the largest perpendicular diameters) at the injection sites. These local responses were associated with reduced mass effect and decreased surrounding edema. Two other patients had stable disease for more than four months (progression-free survival at four months, 9%). The one-year survival rate was 28%. The median survival time for all patients was 7.2 months from the time of enrollment (95% confidence interval, 4.8–12.7 months). Progression-free survival at six months was 4.5%. In conclusion, an independent scientific committee recommended a dosage of 20 mg for a phase 2 clinical trial. This study demonstrated that local treatment with CpG ODNs in patients with recurrent glioblastoma is feasible and well-tolerated at doses up to 20 my [95]. A phase 2 trial evaluating the efficacy of CpG-28 did not achieve the targeted PFS benefit in patients with recurrent GBM. However, the occurrence of a few long-term survivors suggests that some patients with GBM might benefit from this treatment. Translational studies are ongoing to clarify the criteria for the selection of such a subgroup of patients. In addition, a randomized phase II trial is currently ongoing for newly diagnosed GBM, in combination with surgical resection and radiotherapy [96].

A CpG28 phase I trial conducted on patients with different types of cancer, administered alone or concomitantly with oncological treatment, was well tolerated at doses of up to 0.3 mg/kg subcutaneously and 18 mg intratumorally; however, poor effectiveness was observed in glioma patients. There was no significant survival between the groups treated with CpG-28 alone or CpG/oncological therapy. However, the patient with grade III ependymoma was stable during the protocol and remained alive six years after the study. Patients with grade III anaplastic oligoastrocytoma and glioblastoma showed clinical improvement after the administration of CpG28 and bevacizumab. They remained stable for 5 and 8 months and died at 12.5 and 8.8 months, respectively [97].

More recently, a phase II trial indicated that the injection of CpG-ODN into the surgical cavity of newly diagnosed GBM patients after tumor removal was followed by standard of care with radiotherapy and temozolomide. This study enrolled 81 patients who were randomly assigned to receive CpG-ODN plus SOC (39 patients) or SOC alone (42 patients). The incidence of adverse events was similar in both arms, although fever and postoperative hematomas were more frequent in the CpG-ODN arm. This resulted in an increased 2-year survival rate (31% vs. 26%) and median PFS (9 vs. 8.5 months) compared with applying the standard of care alone (NCT00190424) [98].

Although well-tolerated, local immunotherapy with CpG-ODN injected into the surgical cavity after tumor removal followed by SOC did not improve the survival of patients with newly diagnosed GBM [98].

Nevertheless, clinical trials with monotherapy CpG-ODN into glioma tumors, although safe and well-tolerated CpG and showing promise as immunotherapy in mouse models but proved disappointing results in human trials, demonstrated inefficiency in treating patients with glioma. Moreover, it exhibits grade 2 common adverse effects, such as lymphopenia, anemia, neutropenia, local erythema at injection sites, fever, and neurological worsening or fatigue [96,98]. The application of novel combinatorial strategies in clinical trials is of great importance. Preclinical in vitro and in vivo glioma models show an efficient response to the administration of CpG 28 when combined with radiotherapy, with a total remission in two-thirds of animal models [99]. CpGODN1668 induces efficient antitumor immunity in a therapeutic GL261 glioma model [100]. CpGODN 107 also exerts a radiosensitizing effect and induces autophagy and cell cycle arrest, and inhibits angiogenesis [101,102,103]. CpG-1826 combined with cyclophosphamide treatment elicited an antitumor immune effect by a local increase in the main stockholders of the immune system GAMs, DCs, B-cells, and cytotoxic T-cells, favoring an immune memory response and leading to long-term tumor regression [104]. Moreover, the development of nanoparticles has presented a promising strategy to enhance drug delivery and immune response in glioma treatment and thus overcome inefficient therapy. Single-walled carbon nanotubes non-covalently functionalized with CpG SWNT/CpG are the first nanomaterials that inhibit the migration of cancer cells driven by the antioxidant capacity of the SWNTs while stimulating macrophages through induction of the TLR9-NF-κB pathway [105]. Schizophyllan, a polymer that protects short DNA from endosomal degradation, efficiently enhances CpG-ODNs, inducing high levels of inflammatory cytokines to repolarize M2 macrophages to M1 macrophages and induce apoptosis [106]. A synthetic high-density lipoprotein-mimicking nanodisc was reported to deliver CpG with docetaxel (a chemotherapeutic agent) to elicit CD8^+^ T-cell infiltration in glioma models, leading to long-term survival and development of anti-GBM immunological memory when combined with radiotherapy [107]. In vivo, the vaccine (STDENVANT) composed of DCs, CpG-ODN, and GSC lysate as a source of GSC-associated antigens, increases the priming of effector T cells. Furthermore, Combining STDENVANT and anti-PD-L1 antibody diminish Treg in the brain with better survival [108]. (Table 1).

In a recent phase I clinical study of highly antigenic M032 (NSC 733972), Genetically Engineered HSV-1 Expressing IL-12, which contains rich unmethylated CpG in its DNA recognized by TLR9, provoked an adjuvant effect. This trial will be intracranially administered to patients with recurrent/progressive glioblastoma multiforme, anaplastic astrocytoma, or gliosarcoma treated with checkpoint inhibitors after being tested safe in animal models [109].

## 6. TLR9 in Glioma: Dichotomic Role

### 6.1. TLR9 Can Participate in Immune Responses against Glioma

Some studies have revealed that TLR9 plays an important role in glioma progression and induction. For example, Meng et al. (2005) showed that radiotherapy (RT) could be advantageously associated with the intratumoral injection of the TLR9 agonist CpGODN28. When Fisher rats bearing 9 L glioma were treated with various combinations of RT with CpGODN28, complete remission was achieved in two-thirds of patients. The combination of CpG ODN and RT enhanced the role of T cells in antitumor effects [99]. Moreover, another TLR9 agonist, CpGODN107, was investigated as a radiosensitizer in vitro (human glioma U87 cells) and in vivo. The combination of CpGODN107 and irradiation significantly inhibited cell proliferation. The mechanism of radiosensitivity involves CpGODN/TLR9 activation of NF-κB and the production of nitric oxide (NO), inducing cell cycle arrest at the G1 phase but not apoptosis [101]. The radiosensitizing effect of CpG ODN107 is intimately linked to the TLR9-ERK-mTOR signaling pathway. This mechanism is essential for priming autophagic cell death in glioma cells [102]. Inhibition of angiogenesis via the HIFα/VEGF pathway is directly associated with increased CpG ODN107 radiosensitivity in human glioma U87 cells [103]. Moreover, CpG induces apoptosis in glioma cells in vitro and in vivo. It enhances the antigen-presenting capacity of microglia, shifts the immune response to CD8 T cells, and diminishes the number of CD4CD25 Treg [110]. Other studies have also confirmed these results. It has been shown that a unique intratumoral injection of CpG-ODN 1668 successfully abolished glioma growth in vivo and healed 80% of glioma-bearing C57BL/6 mice [100], restoring antitumor immunity in a therapeutic murine glioma model [111]. IGF-1 leading to augmentation of HIF-1α stimulation is coupled with reduced TLR9 and CXCR4 levels and increased SOCS3 expression. These findings suggest a complicated interplay between TLR9 and HIF-1α in response to IGF-1 under normoxia [112].

The safety of intrathecal inoculation with CpG-28 in patients with neoplastic meningitis has also been demonstrated [95,96]. In vitro, the combination of cyclophosphamide (CPA) and CpG-ODN coupled with chemotherapy can elicit antitumor immune responses with reduced length and amount of cure injection [104].

Post incubation with CpG ODN and nanoparticles or encapsulated oligonucleotides (CpG-STAT3ASO) have both antitumor effects by inducing immunostimulatory proprieties and converting M2 scavenging towards beneficial pro-inflammatory type M1 macrophages [106,113,114].

In addition, intravenous and intranasal administration of CpG nano-immunoadjuvant (t-NanoCpG), either alone or with radiotherapy, boosts immunotherapy of glioma by stimulating the maturation of dendritic cells, antigen cross-presentation, and production of pro-inflammatory cytokines in vivo [115].

Based on the information presented by the investigators, TLR9 is an important molecule that can induce or stimulate immune responses against gliomas. These findings point towards the role of CpG in glioma immunotherapy and as a radiosensitizer and provide a rationale for additional clinical advancement of CpG therapy in patients with malignant glioma. Despite these promising results using TLR9 agonists for glioma antitumor therapy [88], CpG-ODN treatment may not yield beneficial effects in glioma patients showing an increase in tumor size after CpG-ODN intratumoral injection in a rat glioma model [116].

### 6.2. TLR9 Promotes Glioma Development

Although TLR9 is involved in the immune response against glioma to eliminate the tumor, its pro-tumoral role in glioma development has been investigated in several studies. Several reports have demonstrated a link between TLR9 expression in human glioblastoma multiforme tissues and patient follow-up. These findings suggest that TLR9 and NFKBIA expression are significant independent prognostic factors for the overall survival of patients with GBM [86,117]. Furthermore, the activation of TLR9 expressed in glioma cells can effectively promote cellular invasion of cancer cells in vitro. CpG oligonucleotides have been shown to stimulate the invasion of U373 astrocytomas overexpressing TLR9 via MMP13 [78]. These results were confirmed by Wang using the U87 glioma cell line [118]. Curiously, a CpG-induced invasion could be abolished by the inhibition of the TLR9 signaling pathway, chloroquine [118,119]. It has been demonstrated that neighboring oxygen levels have an important effect on TLR9 expression and function in human brain cancer cells in vitro. In addition, these studies further suggest a powerful link between TLR9 expression and the invasive machinery in brain cancer cells which can promote brain cancer hypoxia-induced invasion by the activation of MMP 2, 9, and 13 in brain tissues. Thus, TLR9 promotes hypoxia-induced brain cancer cell invasion [80].

TLR9 activation with CpG ODN promotes glioma stem-like cell (GSC) growth. CpG-ODN treatment leads to Frizzled4- JAK2- STAT3 axis activation. In contrast, inhibition of TLR9 abolishes GSC development. These findings point towards choosing TLR9 as a valuable marker for GSC and an immunotherapeutic target for designing effective anti-glioma drugs [92]. Overall, these results suggest that TLR9 suppression and activity inhibition by chloroquine may be used as an adjuvant in GBM therapy, as previously shown by Briceno et al. [120,121].

## 7. Conclusions

For the past several years, glioma treatment has been based on surgery and radiotherapy, but a new understanding of the molecular interaction between the host and the tumor may optimistically lead to interesting developments in this field. Currently, TLR9 has several effects on tumors. On the one hand, TLR9 and its agonist CpG can have beneficial host effects and destroy tumor growth; on the other hand, they can promote tumor progression. To explain these discrepancies, the TLRs function should be studied in detail for each neoplastic setting. Further studies are needed to clarify the dual functions of TLR9 in glioma development and progression (Figure 3). The underlying molecular mechanisms behind TLR9 signaling activation are also needed to better understand the process of these dual effects of TLR9 in gliomas.

## Figures and Tables

**Figure 1 cells-12-00152-f001:**
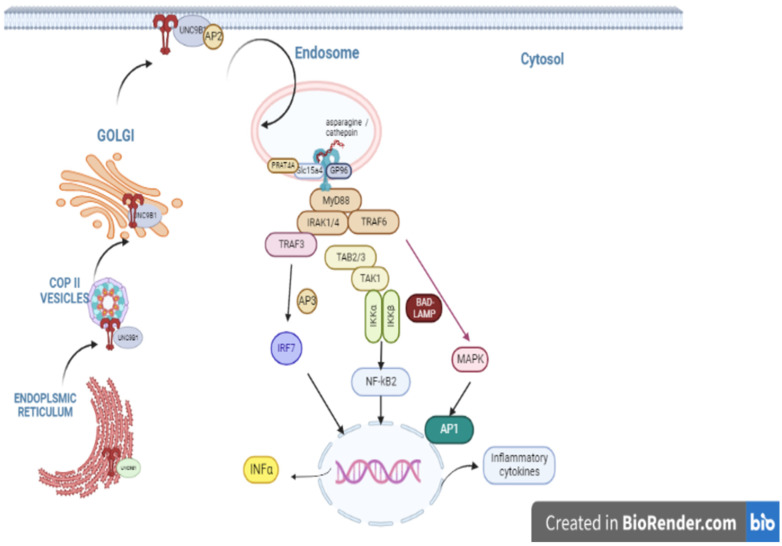
An overview of TLR9 trafficking and signaling pathway. TLR9s are synthesized in the ER to access the cell surface. They require chaperone molecules UNC93B1 to facilitate loading into COPII vesicles and transport to the Golgi. Then, reaching the Surface TLRs Trafficking to endosomes in DCs depends on AP 2, gp96, Slc15a4, and Slc15a4, accessory proteins needed for TLR9 transport. Upon entry into this compartment, TLR9 is proteolytically cleaved by cathepsins and AEP to generate a mature form. It is in this compartment that signaling leads to activation of the NF-κB pathway through (BAD-LAMP/LAMP5), controlling TLR9 trafficking or AP-3 to type I interferon signaling. The activation of TLR9- IRAK- TRAF signaling pathway triggers in turn the stimulation of transcription factors such as mitogen-activated protein kinase (MAPK), activator protein 1 (AP-1), and the nuclear factor NF-Κb. TLR9-IRAK1-TRAF3 signaling cascade leads in turn to type I interferon.

**Figure 2 cells-12-00152-f002:**
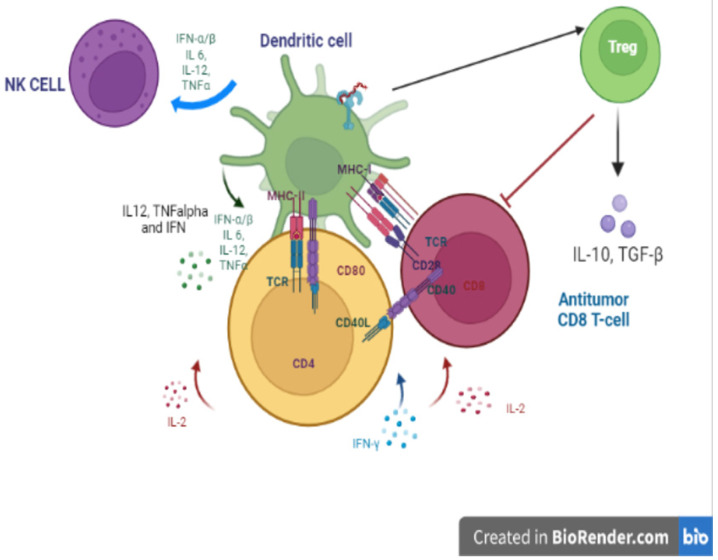
Cross-talk between innate and adaptive immune cells. Upon DC stimulation by CpG-DNA, production of TNF-α/β, IL-6, and IL-12, and upregulation CD40, CD70, B7-1 (CD80), B7-2 (CD86), and major (MHC) class II accrue. Upon antigen capture, immature DCs become licensed and activate naïve T cells. DCs are implicated in the cross-presentation of MHC-I molecules. CD4 T-cell engagement induces surface expression of the CD40 ligand, stimulating CD40 signaling in DC. The costimulatory molecule CD70 is also implicated in the priming of CD8 T. NK cells are strongly activated by CpG-ODNs. Depending on the context of Ag presentation, pDCs lead to immunity by stimulating T-cell priming or promoting the induction of T-cell tolerance by inducing regulatory T cells (Tregs), which promotes tumor progression in several solid tumors.

**Figure 3 cells-12-00152-f003:**
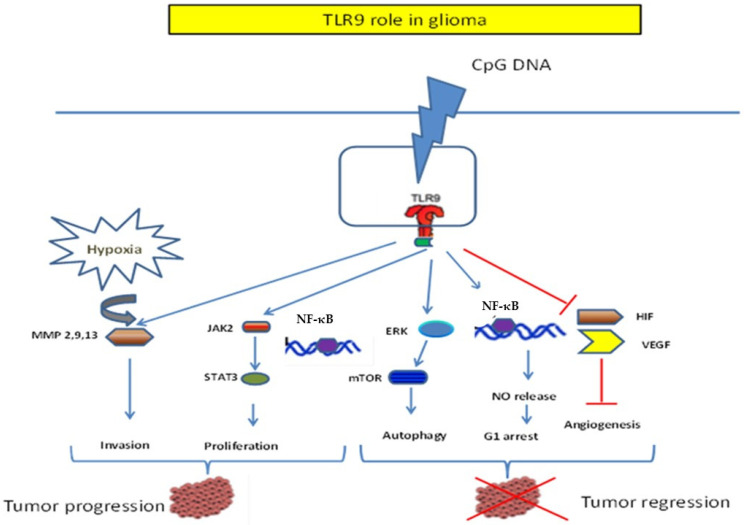
Dichotomic role of TLR9 in Glioma. The role of TLR9 in Glioma on one hand could have an antitumor effect. In fact, TLR9 agonists induce inhibition of angiogenesis by inhibition of VEGF and HIF secretion, inducing G1 arrest of the cell cycle through NF-κB pathway. On the other hand, TLR9 has a dark side, leading to tumor progression mediated through the activation of the JAK2 and STAT3 transcription factor TLR9 mediate CpG oligonucleotide-induced cellular invasion by increasing the matrix metalloproteinase MMP2, 9, and 13 levels through hypoxia.

**Table 1 cells-12-00152-t001:** TLR9 antitumoral based treatment upon stimulation with CpG.

Agostist-TL9	Combinational Treatment	Target	Featured Outcome	Mechanistic Features	References
CpG 28	ns	Fisher rats bearing 9L glioma	complete tumor remission in one-third of the animals	T cells in antitumor effects	[99]
	radiotherapy	Fisher rats bearing 9L glioma	complete tumor remission in two-thirds		[99]
CpGODN1668	ns	glioma-bearing C57BL/6 mice.	inhibit of glioma growth in vivo and cured 80% of animals	diminish Treg and increase CD8	[100]
		murine GL261 glioma cells in vitro	inhibit GL261 cell proliferation		[100]
CpG ODN 107	radiotherapy	U251 and U87/orthotopic tumor-bearing nude mice	induce autophagy	TLR9-ERK-mTOR signaling pathway,	[102]
CpG ODN 107	radiotherapy	U87/human U87 implanted xenographt in nude mice	not induce apoptosis but induce cell cycle arrest at G1 phase/inhibit angiogenesis	TLR9-mediated NF-κB activation and NO production in the tumor cells/VEGF/HIF inhibition	[101,103]
CpG 1826	metronomic cyclophosphamide	GL261 mouse glioma cells/GL261 tumor-bearing mice	elicit anti-tumor immune response	increased tumor T-cell infiltration	[104]
CpG 1826	Schizophyllan (SPG) nanoparticles	C6	repolarizing the M2 macrophages to much-desired M1 and apoptosis		[106]
CpG ODN	carbone nanotubes SWNT	K-Luc murine glioma cell line	inhibit cell migration, activate macrophage	decreased NF-κB activation in glioma cells	[105]
CpG ODN	DTX-sHDL-CpG nanodiscs	Mouse, GL26-WT, GL26-Cit, GL26-OVA, rat CNS-1, and human HF2303, U251	tumor regression and anti-tumor CD8^+^ T-cell responses in the brain TME	long-term survival and immunological memory	[107]
CpG ODN	DTX-sHDL-CpG nanodiscs +RT	of GBM-bearing animals	tumor regression and long-term survival in 80%		[107]
CpG ODN	DCs harboring (GSC)-associated antigens	orthotopic mouse model of glioma	improved survival and tumor regression by enhancing anti-tumor immune function	upregulated programmed death 1 (PD-1) and its ligand PD-L1, decreased T cells	[108]

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
