# Peer review of "TLR9 and Glioma: Friends or Foes?"

_cells, 2022, doi:10.3390/cells12010152_

Round 1
Reviewer 1 Report
Emna et al., reported a review study on TLR9 and Glioma where they discussed the implication of TLR9 in the pathogenesis of glioma. The topic is interesting however the review discussion doesn’t fellow the major points mentioned in the abstract.
1- TLR9/MMP-2/MMP-9 should be developed with more details
2- As TLR9 is expressed more in innate immune cells such as monocyte/macrophages/DC/..It will be better to discuss the role of TLR9 in innate and adoptive immune responses in details to clarify the main target cell population’s dependent on TLR9 signaling.
Minor comments: Abstract line 3 what do you means by “switch immune response towards CD8 T cells”?
manuscript should be revised for few typo errors
Reviewer 2 Report
Authors have tried to elucidate the role of TLR9 in Glioma. TLR9 is a key sensor of CpG DNAs and play a controversial role in cancer. Being Pro-inflammatory itself, TLR9 may play vital role in containing the cancer progression if induction of immune system is desired; nonetheless, overactivation of immune system in a hyperactive tumor-microenvironment leads to an exacerbated situation.
Authors need to get an in depth insights of TLR9 biology and its role in cancer before compiling such study. Refer to the following article for better understanding:
https://doi.org/10.1016/j.semcancer.2019.05.002
10.2147/OTT.S247050
Extensive revision is required. A review article does not require a structural abstract, avoid methods and results headings..
Round 2
Reviewer 1 Report
The paper is suitable for publication from my side.
Reviewer 2 Report
Comments have been addressed.